

# Improvement and generalization of ABCD method with Bayesian inference

Ezequiel Alvarez[1*], Leandro Da Rold[2†], Manuel Szewc[3‡],
Alejandro Szynkman[4°], Santiago A. Tanco[4§] and Tatiana Tarutina[4¶]

**1** International Center for Advanced Studies (ICAS) and ICIFI-CONICET, UNSAM,
25 de Mayo y Francia, CP1650, San Martín, Buenos Aires, Argentina
**2** Centro Atómico Bariloche, Instituto Balseiro and CONICET,
Av. Bustillo 9500, 8400, S.C. de Bariloche, Argentina
**3** Department of Physics, University of Cincinnati,
Cincinnati, Ohio 45221,USA
**4** IFLP, CONICET - Dpto. de Física, Universidad Nacional de La Plata,
C.C. 67, 1900 La Plata, Argentina

⋆ sequi@unsam.edu.ar , † daroldl@ib.edu.ar , ‡ szewcml@ucmail.uc.edu ,
○ szynkman@fisica.unlp.edu.ar , § santiago.tanco@fisica.unlp.edu.ar ,
¶ tarutina@fisica.unlp.edu.ar

## Abstract

To find New Physics or to refine our knowledge of the Standard Model at the LHC is an enterprise that involves many factors, such as the capabilities and the performance of the accelerator and detectors, the use and exploitation of the available information, the design of search strategies and observables, as well as the proposal of new models. We focus on the use of the information and pour our effort in re-thinking the usual data-driven ABCD method to improve it and to generalize it using Bayesian Machine Learning techniques and tools. We propose that a dataset consisting of a signal and many backgrounds is well described through a mixture model. Signal, backgrounds and their relative fractions in the sample can be well extracted by exploiting the prior knowledge and the dependence between the different observables at the event-by-event level with Bayesian tools. We show how, in contrast to the ABCD method, one can take advantage of understanding some properties of the different backgrounds and of having more than two independent observables to measure in each event. In addition, instead of regions defined through hard cuts, the Bayesian framework uses the information of continuous distribution to obtain soft-assignments of the events which are statistically more robust. To compare both methods we use a toy problem inspired by $pp \rightarrow hh \rightarrow b\bar{b}b\bar{b}$, selecting a reduced and simplified number of processes and analysing the flavor of the four jets and the invariant mass of the jet-pairs, modeled with simplified distributions. Taking advantage of all this information, and starting from a combination of biased and agnostic priors, leads us to a very good posterior once we use the Bayesian framework to exploit the data and the mutual information of the observables at the event-by-event level. We show how, in this simplified model, the Bayesian framework outperforms the ABCD method sensitivity in obtaining the signal fraction in scenarios with 1% and 0.5% true signal fractions in the dataset. We also show that the method is robust against the absence of signal. We discuss potential prospects for taking this Bayesian data-driven paradigm into more realistic scenarios.

## Contents

## 1 Introduction

Given the culmination of the extremely successful program of the Large Hadron Collider (LHC) on the horizon, the expected increase in luminosity by a factor of about ten, and the lack of significant excesses in recent LHC analyses, it becomes compelling to focus on developing new strategies that go beyond the accumulation of statistics in the task of finding New Physics or establishing better measurements in Standard Model observables. There are many reasons leading the community on that course: it is well known that we are continuously learning and understanding more about the LHC and its detectors, Monte Carlo simulations used for predictions are improving their accuracy, new observable and analysis techniques are being constantly designed and developed. In this article we focus on the latter of these, since it could provide fertile room for progress and advancement. We study improvements for data-driven techniques, which in particular are especially useful for signals with few expected events.

Data-driven techniques are very important at the LHC, since in many cases the expected backgrounds and the signal are difficult –if not impossible– to model and simulate at a reliable level of unbiasedness and precision. In addition, data-driven techniques are a robust complement to other procedures, and many times represent a confirmation for them. Maybe the most simple data-driven technique is the measurement of a resonance in some invariant-mass (or any other) distribution, since in this case one is doing side-band fitting to a curve and a significant excess at some point indicates the resonance. The strategy that led to the Higgs discovery was much along these lines.

A more involved, but still brilliantly simple, data-driven modelling technique is the ABCD method [1–4]. This approach consists in finding two independent observables that classify the background into four regions, such that the signal lies only in one of them. The method allows to predict the signal events in its corresponding region as long as its underlying hypotheses are satisfied. The method has been successfully implemented by the community and it can be found in many of the ATLAS [5] and CMS [6] public results, with some particular examples listed in Refs. [7–11]. In many applications, the method is adapted or extended to the particular features of the observable and backgrounds. The ABCD approach is quite useful

because it reduces the impact of potential biases and inaccuracies in simulations, and because of its simplicity it only relies in finding the observables and regions that fulfill the required assumptions.

The ABCD method has been studied in the literature and it has received a variety of proposals for improvements. In Ref. [12] a set of extended ABCD methods is suggested to tackle the problem of observables with a slight dependence between them. Ref. [13] consists of a detailed guide for applying the ABCD method, and in which signal contamination in control regions is discussed using a matrix method. Ref. [14] showcases how the independent observables used in ABCD could be designed through Machine Learning techniques instead of choosing them through first principle arguments or physical intuition. This is a very appealing proposal that uses Machine Learning to construct the independent observables through a loss function that should be fed with labeled pseudo-data generated through simulations. Although one may find ambiguous to use simulations to enhance a method whose idea is to reduce the impact of simulations, the increase in statistical power warrants further exploration.

In this work we propose to study an existing method from the area of Statistics, more specifically from Bayesian Machine Learning, that consists in disentangling many classes which are mixed in a given dataset. This procedure, known as a *mixture model* [15], considers that the dataset $\mathbf{X}$ is a realization of a probability density function for $K$ classes with class probabilities $\pi_k$ and per-class probability densities with parameters $\theta_k$. Using the Bayes Theorem, the method yields –most of the times numerically– a probability distribution for the parameters of the model, $p(\theta|\mathbf{X})$, given a collection of data $\mathbf{X}$, namely events at LHC, each containing the value of several observables measured in the same event. This distribution is called the posterior of the model. In contrast to ABCD, observables in the mixture model can have continuous distributions that, by definition, contain more information than discrete outputs. A mixture model as presented here assumes that each data point (event) belongs to only one of the possible classes.

The mixture model has among its parameters the expected fraction of each class in the dataset, and in particular for the class that corresponds to the signal, this fraction yields the expected amount of signal in the dataset, which is generally the sought-after unknown. In addition, the Bayesian inference also finds the posterior distribution for all the other parameters as well. This posterior refines the understanding of the physical system, since most of the posterior parameters have a connection to the physics involved in the problem. Moreover, as it can be seen from the mathematical formulation of the mixture model, the number of classes is not fixed (while in the ABCD method there can only be signal and background), and the number of observables of each data point is also not limited to any number (while the ABCD method is restricted to two independent observables). For the purposes of simplicity and performance we also demand independence for the observables measured in each data point in the mixture model.[1] Another recognizable feature of the mixture models is that they do not need control regions, and that there are no hard cuts to define regions nor hard-assignments to define whether a given event belongs to a specific class. Instead, each event is assigned a probability of belonging to each class.

In our understanding, one of the most important advantages of the presented tool and framework, is that it facilitates the full exploitation of the existing dependence between the observables at the event-by-event level. Or in more statistical terms, it fully exploits the mutual information in the multi-dimensionality of the data. In this article we aim to explore an understanding of up to what extent such a paradigm could improve sensitivity with LHC observables.

---

[1]A more complex scenario with dependent observables could also be tackled through a mixture model as long as one has some prior-knowledge or clues about this dependence. We do not address this possibility in this work.

To compare both methods we use a toy problem whose basic properties are inspired by the di-Higgs production at LHC, with both Higgs states decaying to $b\bar{b}$, namely: $pp \rightarrow hh \rightarrow b\bar{b}b\bar{b}$. This process, being one of the best candidates to probe the nature of the Higgs boson, and one of the most relevant measurements to be expected by the LHC in the forthcoming years, has been studied in [10, 11]. In these works one can find that the ABCD method with some modifications is utilized as part of the pipeline for extracting the results. The process has several interesting features, such as a small cross-section, large backgrounds and, given the class, the presence of several independent or approximately independent observables, such as the four $b$-tagging scores[2] of the jets and the two invariant masses of the corresponding paired jets. As a first step in the analysis of this process with Bayesian inference, we consider in this article a toy model that keeps some of its properties, as well as just two of its backgrounds, and analyse it with the ABCD method and with Bayesian inference. We study and quantify both performances and provide details on why the Bayesian analysis has very good perspectives.

The developments and results in this work are a proof-of-concept and are still far away from an application in a realistic scenario. In any case we provide a brief discussion about it in Section 4. Finally, it should be mentioned that the results in this work consist mainly of importing, adapting and discussing tools, techniques, skills and algorithms from Statistics and Bayesian Machine Learning industry to LHC physics.

This work is divided as follows: in Section 2 we discuss some probabilistic models for the analysis of data at LHC, presenting an overview of the ABCD method in 2.1 and introducing Bayes inference techniques in Section 2.2. As a connection between both methods we show that the mixture model, although being a different paradigm than the ABCD method, can be reduced to the latter. We also study how it improves and generalizes its performance and range of applicability. In Section 3 we explicitly compare both methods performance in a toy problem inspired by $pp \rightarrow hh \rightarrow b\bar{b}b\bar{b}$. We take a few benchmark results and compare and discuss the performance of both methods. Section 4 holds a brief discussion of certain details that emerge from the previous sections results, as well as some milestones that should be achieved in order to take the presented idea to production. In Section 5 we present the conclusions of the work.

## 2 From ABCD to probabilistic models for data-driven LHC analyses

In this Section we frame probabilistic models as a natural extension of data-driven techniques in High Energy Physics with the ABCD method as a starting point. In 2.1 we present an overview of the ABCD method as currently used in most applications and in 2.2 we show how probabilistic mixture models can be used to improve upon the ABCD method by relaxing certain hypotheses at the expense of other modelling assumptions.

### 2.1 The ABCD method: An overview

The ABCD method, as it is known and used in High Energy Physics [7–11], is an established data-driven method to estimate background in a signal region. Therefore, by counting the total number of events in this region, one can also estimate the number of signal events in the signal region which is usually the main objective in many analyses.

The method consists in finding two observables which have independent distributions for the background and that can be simultaneously measured in each event, say $\mathcal{O}_1$ and $\mathcal{O}_2$. We then divide the outcome of each one of the two observables into two regions, in such a way that the signal lies in only one of the four regions that are determined by the aforementioned

---

[2]There are systematic sources that may yield a small dependence between these scores. Although we ignore this dependence in the toy problem studied along is work, it should be taken into account in a more realistic study.

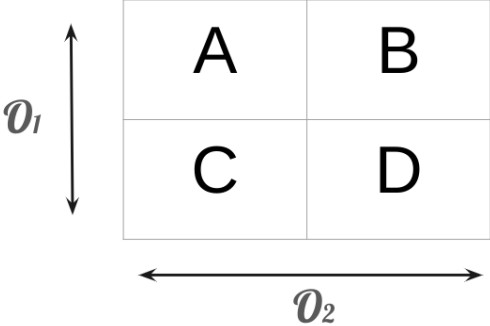

Figure 1: ABCD method: observable $\mathcal{O}_1$ can take values which are either AB or CD. Whereas observable $\mathcal{O}_2$ can only take values which are either AC or BD. Assuming that signal is restricted to A, and that the $\mathcal{O}_{1,2}$ distributions for the background are independent, one has that $N_A(\text{background}) = N_B \times N_C / N_D$, see text for details.

division. These regions are usually named A, B, C and D. For the sake of clarity we define the region e.g. AB as the one containing all the events in A and all the events in B; and so on with all the combinations. Therefore, we can name the regions such that the output for $\mathcal{O}_1$ is either in the region AB or CD, and the output for $\mathcal{O}_2$ is either in AC or BD, as in Fig. 1. Henceforth, if we make the choices such that signal only lies in region A, then it is easy to show that under the mentioned assumptions the number of background events in this region will be determined through

$$N_A(\text{background}) = N_B \times N_C / N_D \,. \tag{1}$$

Where $N_X$ refers to the number of events in region $X$. $N_A(\text{background})$ refers to the number of background events in region $A$. This result is easily understood because if the events are distributed according to the background distribution in which the observables are independent, then one expects the ratio $N_C / N_D$ to be equal to $N_A(\text{background})/N_B$. Observe that we assume that only region A contains signal. In the literature one can sometimes find that B is named as control region, and $N_C / N_D$ as a transfer function.

As it can be appraised, the ABCD method is very clever and simple, and it has worked with excellent results and achievements in HEP, as discussed in the introduction. However, if the hypotheses are not exactly satisfied, the predictions would deviate from their corresponding true values, as expected for any statistical analysis with the same hypotheses. Not only that, but also if the total number of events in B, C or D is small, then its Poisson fluctuation would propagate to the signal and background events predicted in A, a characteristic which is potentially alleviated in our proposed method by using differential information.

Therefore, it is also interesting to analyze some limitations in applying the ABCD method. As a first observation one should notice that the method uses *hard cuts* to define the regions; whereas mixture models can deal with all the information contained in the dataset. Using this information, applications in statistics have shown improvements when using *soft-assignments* [16]. That is, each event has a given probability of being signal and a given probability of being background (in mixture models this is usually referred as *responsibility*). As a second point, one should notice that the ABCD method is limited to two independent observables. Because of this, when there are more independent observables then some of them must be combined to reduce the number of observables, and/or the method is *binned* in the other observables. In the latter scenario, one computes many ABCD methods according to the binned output of the other observable(s). Also, if it is the case that there are many different backgrounds, then the ABCD method does not have the tools to exploit this knowledge beyond its formula in Eq. 1.

## 2.2 Improving and generalizing ABCD through Bayes inference techniques

Here we tackle the limitations mentioned above, and we show how to improve and generalize the ABCD method through standard Bayesian inference techniques. It is *improved* because the hard cuts are improved upon by incorporating all the information contained in the observables through the soft-assignments. It is *generalized* because the new framework allows to consider simultaneously many independent observables, and because it also allows to exploit the knowledge of many different backgrounds. And finally, it is an improvement and generalization of *the ABCD method* because the latter is a special case of the former, as shown at the end of this section. The proposed framework is a probabilistic model which is described in the following paragraphs.

From the physical point of view, the dataset is a given selection of $N$ events in which $D$ independent observables have been measured in each one of the events. We assume that each event corresponds to one of $K$ possible processes that pass the selection criteria. Within these $K$ possible process there are $K-1$ backgrounds and the signal.

We model this dataset from the mathematical point of view as the outcome of a probabilistic generative model depicted in the Graphical Model [15] in Fig. 2. Graphical models are a convenient way to depict probabilistic generative models as they allow for clear interpretability but also serve as a bookkeeping technique that can be useful when performing inference. We assume that, given the $n^{th}$ event, its class is determined through the sampling of a categorical latent random variable $\mathbf{z_n}$, which can take $K$ values and is not observed. Then, the $D$ observables in this event are sampled using the distributions from the class corresponding to the value of $\mathbf{z_n}$, and we assume that each observable has no dependence on the other $D-1$ observables once the class has been sampled. Henceforth, although $\mathbf{z_n}$ is not observed, the conditionally independent outcome of the $D$ observables provides information about the probability of the event to belong to each one of the classes. Moreover, the $D$ observables also provide information on their own distributions within each class. Bayesian inference provides a framework to exploit all this data.

In standard inference techniques such as approximate Variational Inference or exact Monte Carlo techniques [15], it is customary to use a 1-of-K representation for the latent variable $\mathbf{z}_n$, expressing the variable as a $K$ dimensional vector indexed with only one non-null entry, when writing down the corresponding probability density associated with the latent variable and the corresponding data point $\mathbf{x}_n$. This implies that the latent variable for a given event is a vector with only one component equal to 1 and all other components 0. With this notation the probability for the $n^{th}$ data point becomes

$$p(\mathbf{x}_n, \mathbf{z}_n | \theta) = p(\mathbf{z}_n | \pi) \prod_{k=1}^{K} \left( \prod_{d=1}^{D} p(x_{nd} | \theta_{kd}) \right)^{z_{n_k}}, \tag{2}$$

where $z_{n_k}$ is the $k^{th}$ component of $\mathbf{z}_n$ and $\theta$ is a vector containing all the parameters of the model which can be split into the class fraction parameters $\pi = \pi_{k=1,...K}$ and the parameters of the distributions for each of the $D$ random variables for the $k$-th class, $\theta_{kd}$, which can themselves be a vector depending on the choice of per class and per observable distribution. To shorten the notation, we will refer to the class-dependent distributions simply as $p(\mathbf{x}_n | k)$. Having represented $\mathbf{z}_n$ as a 1-of-K vector, we observe how the product selects only the class with a non-zero value of $z_{n_k}$. Additionally, $\mathbf{z_n}$ follows a Categorical[3] distribution with the same form

$$p(\mathbf{z}_n | \pi) = \prod_{k=1}^{K} \pi_k^{z_{n_k}}. \tag{3}$$

---

[3]A categorical distribution is the usual multinomial distribution specialized to the case where there is only one draw.

Since $\mathbf{z}_n$ is not observed, we should marginalize over it:

$$p(\mathbf{x}_n|\theta) = \sum_{\mathbf{z}_n = \delta_{kk'}} p(\mathbf{x}_n, \mathbf{z}_n|\theta), \tag{4}$$

where with $\delta_{kk'}$ we mean again a 1-of-K representation, where $\mathbf{z}_n$ is a $K$ dimensional vector indexed by $k'$ with only one non-null entry at index $k$. Because each specific value of $\mathbf{z}_n$ selects one class $k$, the marginalized probability distribution takes the usual mixture model form

$$
\begin{aligned}
p(\mathbf{x}_n|\theta) = \sum_{\mathbf{z} = \delta_{k,k'}} p(\mathbf{x}_n, \mathbf{z}_n|\theta) &= \sum_k p(\mathbf{z}_n = \delta_{k,k'}|\pi) \prod_{k'=1}^{K} p(\mathbf{x}_n|k')^{\delta_{k,k'}} \\
&= \sum_{k=1}^{K} \prod_{k'=1}^{K} \left( \pi_{k'} p(\mathbf{x}_n|k') \right)^{\delta_{k,k'}} \\
&= \sum_{k=1}^{K} \pi_k\, p(\mathbf{x}_n|k).
\end{aligned} \tag{5}
$$

Therefore, the probability for the dataset $\mathbf{X} = \{\mathbf{x}_1, ..., \mathbf{x}_N\}$ is simply the product of Eq. 5 for each data point. This yields

$$p(\mathbf{X}|\theta) = \prod_{n=1}^{N} p(\mathbf{x}_n|\theta), \tag{6}$$

which is one of the fundamental components to run Bayesian inference. We need to add a prior probability for the parameters of the model, $p(\theta)$, and one can in principle obtain the posterior $p(\theta|\mathbf{X})$ through Bayes' theorem

$$p(\theta|\mathbf{X}) = \frac{p(\mathbf{X}|\theta)p(\theta)}{\int d\theta'\, p(\mathbf{X}|\theta')p(\theta')}. \tag{7}$$

This usually can be achieved through a variety of numerical techniques, we describe in Section 3 the tools that we have used along this work. The general methodology can then be summarized as a choice of observables $\mathbf{x}$ and the modelling of the associated likelihood $p(\mathbf{x}|\theta)$ whose parameters of interest are assigned a prior $p(\theta)$. The method proceeds to extract the posterior distribution, or for simplicity the Maximum a Posteriori (MAP) estimates of the parameters, from the measured unlabeled data $\mathbf{X}$, which in turn provides us with estimators of the quantities of interest chief among them the expected number of signal events.

To make the connection between the mathematical framework and the physical problem, we identify $\pi_k$ as the component of the corresponding class $k$ in the mixture, and its posterior is the probability distribution for its fraction in the dataset. Therefore the signal fraction is estimated through its corresponding posterior. Moreover, one can compute the posterior probability of the $n^{th}$ event to belong to each one of the $K$ classes by studying the posterior for $\mathbf{z}_n$. This yields a *soft-assignment* for each event, since it has a probability of belonging to each one of the classes.

One can summarize the contrast in some of the main features from both methods through the following table.

| ABCD | Bayesian framework |
|---|---|
| 2 independent observables. | $D$ independent observables, with $D$ arbitrary. |
| 2 components, signal and background. | $K$ components can make up the mixture model, the signal and $K-1$ backgrounds. $K$ arbitrary |
| Prior knowledge on signal and background distributions allows to define the four hard cut regions A, B, C and D, and to assume that the signal is only found in A. Then the method estimates how many signal events are expected in A. | Instead of regions there is prior knowledge on the distribution of the $K$ components over each one of the $D$ independent observables. Then the data and its mutual information at the event-by-event level provides the information to infer and learn <br><br> • A posterior on the class fractions, and in particular the posterior probability distribution for the signal fraction in the sample. <br><br> • The components distributions over the $D$ observables, whose posteriors are expected to be closer to their true values than its corresponding priors. |
| Signal should be bounded to a region in phase space, and control regions are needed. | Signal and background can be mixed in different proportions in all phase space. There is no need of a control region. |

It is interesting to observe, in any case, that both methods need independent observables.

One of the objectives of this work is to show using an explicit toy problem that the Bayesian method performs better than the ABCD method. This is presented in Section 3.

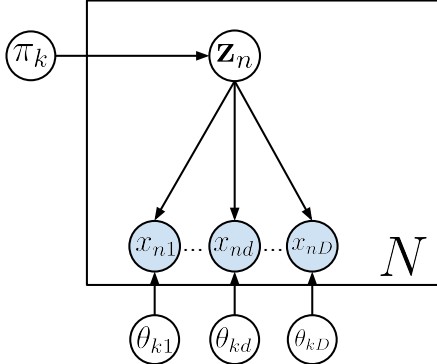

Figure 2: General *Graphical Model* for a mixture model. $k$ runs over the $K$ classes, $n$ runs over the $N$ events, and $d$ over the $D$ independent observables. Random variables are represented by circles while arrows represent conditional dependence. White circles represent latent variables which are unobserved while blue circles represent measured variables whose observation conditions the posterior distribution over the parameters. See, for instance, Chapter 8 in Ref. [15] for details about representing a probability density function using a Graphical Model.

**Recovering ABCD as a special case in the Bayesian inference framework**

We end this section by showing that the starting point for the Bayesian framework, Eq. 5, recovers the ABCD method when the corresponding conditions are fulfilled.

We assume then exactly two observables $\mathcal{O}_{1,2}$ and two classes, namely signal ($s$) and background ($b$). In a hard-cut scenario, the observables outputs are simply $\mathcal{O}_1 = AB$ or $CD$, and $\mathcal{O}_2 = AC$ or $BD$ (see Fig. 1), and thus $p(\mathbf{x}|k)$ is a Categorical distribution for $\mathbf{x}$ which now consists of four possible results $(A, B, C, D)$ and $\theta_k$ is a vector of four probabilities which sum up to one. Therefore, we can write

$$p(\mathbf{x}_n = A|s + b) = \pi_b\, p(\mathbf{x}_n = A|b) + \pi_s\, p(\mathbf{x}_n = A|s) = \pi_b\, \theta_{bA} + \pi_s\, \theta_{sA}, \tag{8}$$

and so on replacing A $\to$ B, C or D. Because of the assumption that signal lies only in region A, the second term is non-zero only for region A. Moreover, in that case $p(\mathbf{x}_n = A|s) = \theta_{sA} = 1$. If we now use the maximum likelihood estimators $p(\mathbf{x}_n = A|s + b) = N_A/N$ (replacing A $\to$ B, C or D) and the fact that the two observables are independent, we can write the four expressions coming from Eq. 8 as the following set of equations

$$N_A = N\, \pi_b\, p(\mathcal{O}_1 = AB|b)\, p(\mathcal{O}_2 = AC|b) + N\, \pi_s\,, \tag{9}$$

$$N_B = N\, \pi_b\, p(\mathcal{O}_1 = AB|b)\, p(\mathcal{O}_2 = BD|b)\,, \tag{10}$$

$$N_C = N\, \pi_b\, p(\mathcal{O}_1 = CD|b)\, p(\mathcal{O}_2 = AC|b)\,, \tag{11}$$

$$N_D = N\, \pi_b\, p(\mathcal{O}_1 = CD|b)\, p(\mathcal{O}_2 = BD|b)\,. \tag{12}$$

From here it is straightforward to get that

$$N_B \times N_C / N_D = N\, \pi_b\, p(\mathcal{O}_1 = AB|b)\, p(\mathcal{O}_2 = AC|b)\,, \tag{13}$$

where the right-hand-side is exactly the first term in Eq. 9 and indicates the expected background in region A, $N_A$(background). Therefore this expression matches the data-driven background expectation in region A according to the ABCD method, Eq. 1, as we wanted to show.

# 3 A $hh \to b\bar{b}b\bar{b}$-inspired toy problem

To exemplify the use of probabilistic models and their differences with traditional ABCD data-driven background estimation as detailed in Section 2, we devise a toy problem that appropriately captures the relevant physics. In the selection of the toy problem, we take inspiration in one of the most relevant LHC benchmarks, di-Higgs production searches in the $4b$ final state. As detailed in Section 1, di-Higgs measurements are one of the most important future measurements accessible to the HL-LHC. Given the challenges of such a measurement, ingenuity will be needed to take full advantage of the available data. We show here how probabilistic models could address some of the drawbacks of current data-driven methods without increasing the dependency on Monte Carlo simulations.

Inspired by the di-Higgs measurement analysis, we consider a simplified toy problem where we have $K = 3$ classes. Two of these classes will be the backgrounds $\mathfrak{b}_1$ and $\mathfrak{b}_2$ which are inspired by two of the main backgrounds on di-Higgs: non-resonant $4c$ and $2b2c$, respectively. The remaining class will be the signal $\mathfrak{s}$, inspired by the di-Higgs production signal. We do not consider backgrounds which are inspired by non-resonant $4b$, single Higgs production or light jets. A realistic analysis of di-Higgs production certainly requires their inclusion.

Having determined the number of possible classes, we need to define the observables that will be used for our probabilistic model. We consider a set of $N$ events, each of which will consist of $D = 6$ measured observables that take inspiration on useful information derived

from the four-jet final state in di-Higgs measurements. These six observables consist of what we call four $b$-tag scores $\mathcal{S}_{i=1,\dots4}$ and two invariant masses $m_{1,2}$. In this work, we assume the jet scores and the invariant masses are all conditionally independent.[4]

The $b$-tag score $\mathcal{S}$ for each of the four jets is a real number whose distribution is bounded. In our toy problem, we assume that we only have two possible types of jets, $b$- and $c$-jets, meaning that each of the four scores in each event can be drawn either from the $b$- or $c$-jet score probability distribution function (pdf). Each class will be distinguished by its jet composition that dictates which probability distribution each of the four scores is drawn from. For $\mathfrak{b}_1$, all four jets are sampled from the $c$-jet pdf; for $\mathfrak{b}_2$ two jets are sampled from the $c$-jet pdf and two from the $b$-jet pdf; and for $\mathfrak{s}$ all four jets are sampled from the $b$-jet pdf. One should note that for $\mathfrak{b}_2$, because there are two true $b$-jets and two true $c$-jets, the drawing of the scores and the resulting probabilistic model is slightly more involved. We detail this in 3.1.

The remaining two observables consist of what we call two invariant masses. These are aimed to replicate what is observed in di-Higgs searches after grouping the four-jets in pairs by minimizing a merit function such as

$$\chi = \sqrt{(m_1 - m_{h_1})^2 + (m_2 - m_{h_2})^2},\tag{14}$$

where $m_{h_{1,2}}$ are the two Higgs masses. We do not model said replication but instead take inspiration from the resulting invariant masses. For simplification, we assume that each mass is sampled either from a resonant or a non-resonant distribution. The only difference between classes again arises by the selection of which pdf the masses follow. For $\mathfrak{b}_1$ and $\mathfrak{b}_2$, the two masses are sampled from the same non-resonant (NR) distribution, while for the signal $\mathfrak{s}$ the two masses are sampled from the same resonant (R) distribution.

Summarizing, the toy problem consists of defining $p(\mathcal{S}_1, \mathcal{S}_2, \mathcal{S}_3, \mathcal{S}_4, m_1, m_2 | k)$ in terms of $p(\mathcal{S}|b)$, $p(\mathcal{S}|c)$, $p(m|R)$ and $p(m|NR)$ and the jet type probabilities for the four jets $p(jjjj = bbbb|k)$, $p(jjjj = bbcc|k)$, $p(jjjj = bccb|k)$,... . We then learn from a set of $N$ events with $D = 6$ observables these underlying tagger efficiency curves, mass distributions and overall class fractions for the $K = 3$ specified processes that we assume are present in the data. To do this, we need to make specific assumptions regarding the parametric forms of the pdfs. For this simplified toy problem, we assume that the four needed probability distributions ($b$- and $c$-jet score pdf and resonant and non-resonant mass pdfs) to sample the 6 observables per event are known, simple parametric functions. As we discuss in Section 4, this is a working assumption that needs to be improved upon to better capture the physics of di-Higgs searches. We detail the relevant parametric functions of the probabilistic model in Section 3.1. Observe that we assume as known the parametric functions, but not their true value, which we infer.

This toy problem is meant to showcase the power of Bayesian inference. There is a higher dimensionality than what is allowed in the standard ABCD, at the expense of additional modelling in specifying the mass and score parametric forms. One could worry as well about overparameterizing the problem. However, the assumption of conditional independence between $b$-tag scores and di-jet masses and the fact that the score pdfs are shared among the processes mitigate this risk. The former assumption is consistent with the ABCD method as detailed in Section 2 while the latter means that each class has the same two available score distributions but uses them differently depending on its jet composition. This composition is not inferred but fixed a priori when deciding which processes are assumed to be present and corresponds to specifying the model appropriately.

For this toy problem, we assume that the model is correctly specified. That is, that the data follows the model with a specific choice of parameters. Although this will not be a realistic assumption if dealing with physical measurements, this allows us to perform a closure test

---

[4]There is a slight subtlety here in that we assume that all variables are conditionally independent only after we specify the jet-type of each jet.

and more importantly a demonstration of the power of the method. We thus can generate pseudo-data according to a specific choice of parameters of the model and perform Variational Inference (VI) [17] with pyro [18] in said pseudo-data to obtain the Maximum a Posteriori (MAP) parameters of the model. VI learns the MAP parameters by approximating the posterior with a mean field function $q(\theta, \mathbf{Z}) = \delta(\theta - \theta_{\text{MAP}})q(\mathbf{Z}|\theta)$, where $\mathbf{Z}$ is the set of all hidden latent variables such as class and jet-type assignment. The $\theta_{\text{MAP}}$ values are obtained by maximizing the Evidence Lower Bound (ELBO) between the posterior proxy $q$ and the joint distribution $p(\mathbf{X}, \theta, \mathbf{Z})$, which is equivalent to minimizing the Kullback-Leibler divergence between the true posterior and its approximation. When VI is used in this manner, it yields similar results to a global fit with the prior acting as a regulator. However, VI can be used with more general approximation functions that retain the probabilistic nature of the posterior, see e.g. [19]. We leave this full posterior estimation for future work and consider here just point estimates. This is both numerically easier but also enough if our main goal is to compare directly with the ABCD method. However, one should keep in mind that computing point estimates instead of posterior distributions undersells the power of Bayesian techniques.

We detail the probabilistic model in Section 3.1 and the results from our Maximum a Posteriori (MAP) parameter inference in Section 3.2.

## 3.1 The toy model for the toy problem

As mentioned above, specifying the toy model implies defining the score probability distributions for each jet type and the possible classes in terms of their jet type composition and mass distributions. We assume that we have two jet types, $b$- and $c$-jets, and that we have three processes, the signal $\mathfrak{s}$ producing a doubly-resonant $4b$ signature and the backgrounds $\mathfrak{b}_1$ and $\mathfrak{b}_2$ that consist in non-resonant $4c$ and $2b2c$ events respectively.

The jet types determine the individual score probability distributions, $p(\mathcal{S}|j)$ with $j = b$- or $c$-jets. Guided by the $b$-tag score distributions in Ref. [20], and since these can be thought as acceptance probabilities, a reasonable and simplistic assumption for this proof-of-principle is to assume a Beta distribution with parameters $\alpha, \beta$ for each jet type,

$$p(\mathcal{S}|j) = \text{Beta}(\mathcal{S}; \alpha_j, \beta_j). \tag{15}$$

The Beta distributions render inference smoother, and we consider Gaussian priors for each parameter. Although the modelling of the distributions with Beta functions is too restrictive for realistic tagging score distributions, it suffices for the proof-of-principle.

As stated in previous paragraphs, each class has its own different possible four jet states. The relevant sample space for four jet states is $bbbb$, $bbbc$, $bbcb$, etc. As detailed above, inspired by the di-Higgs signal and two of its main backgrounds, we assume that the classes are: $\mathfrak{s} = bbbb$, $\mathfrak{b}_1 = cccc$ and $\mathfrak{b}_2 = ccbb$ (in any order). We can phrase the simpler cases as $p(\mathcal{S}_1, \mathcal{S}_2, \mathcal{S}_3, \mathcal{S}_4|\mathfrak{s}) = \prod_{i=1}^{4} p(\mathcal{S}_i|b)$ and $p(\mathcal{S}_1, \mathcal{S}_2, \mathcal{S}_3, \mathcal{S}_4|\mathfrak{b}_1) = \prod_{i=1}^{4} p(\mathcal{S}_i|c)$. For $\mathfrak{b}_2$ the situation is more complicated. Defining the sample space as $\{bbcc, bccb, cbcb, ccbb, bcbc, cbbc\}$, we can define a 1-in-6 latent variable $\mathbf{a}_j$ that encodes which of the six configurations is selected for a given event belonging to the $\mathfrak{b}_2$ class. Because all possibilities are equivalent, the probability for any one of them is $1/6$. We thus model the score pdfs for the $\mathfrak{b}_2$ class as

$$p(\mathcal{S}_1, ..., \mathcal{S}_4|\mathfrak{b}_2) = \sum_{j=1}^{6} p(\mathbf{a}_j)p(\mathcal{S}_1, ..., \mathcal{S}_4|\mathfrak{b}_2, \mathbf{a}_j)$$

$$= \frac{1}{6}\left(p(\mathcal{S}_1|b)p(\mathcal{S}_2|b)p(\mathcal{S}_3|c)p(\mathcal{S}_4|c) + \text{permutations}\right). \tag{16}$$

When running inference on our probabilistic model, we consider the joint distribution $p(\mathcal{S}_1, ..., \mathcal{S}_4, \mathbf{a}_j|\mathfrak{b}_2)$ and thus sample $\mathbf{a}_j$ as well using a Categorical distribution with probabilities $\frac{1}{6}$.

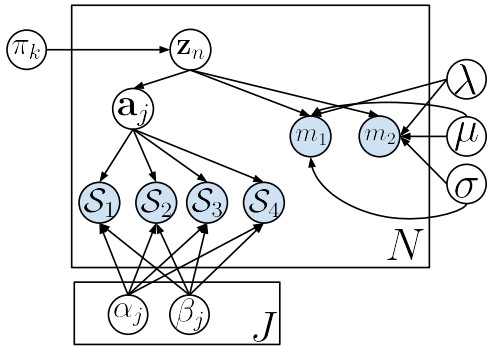

Figure 3: Graphical Model for the probabilistic model considered for our toy problem. Observation of the six-dimensional data consisting of four $b$-tag scores $\mathcal{S}_{1..4}$ and the two invariant masses $m_{1,2}$, conditions the posterior distribution of the parameters of interest $\theta = \{\pi_k, \alpha_j, \beta_j, \lambda, \mu, \sigma\}$. Here $N$ runs over the events and $J$ runs over the two individual types for jet classification, c- and b-jets. Prior hyperparameters not shown here are specified in the text.

With regards to the mass distribution, we assume that the backgrounds are non-resonant and thus the mass distribution for both measured masses is an exponential

$$p(m|\text{NR}) = \text{Exponential}(m; \lambda), \tag{17}$$

where $\lambda$ is the decay rate. We emphasize here that we are making a further simplifying assumption: because the mass distribution for the non-resonant backgrounds is assumed to be independent of jet types, we can consider the same exponential distribution for both processes. That is, if the process is non-resonant, then the distribution is determined by a shared parameter $\lambda$ for all such classes. We consider a uniform prior for $\lambda$ when performing inference.

The signal on the other hand will be resonant, with

$$p(m|\text{R}) = \mathcal{N}(m; \mu, \sigma), \tag{18}$$

where $\mu, \sigma$ are the usual mean and standard deviation of the normal distribution $\mathcal{N}$, and we assume normal priors for them as well. To mimic realistic preselection cuts, we consider a fixed mass window of $[75, 175]$ GeV and modify the mass distributions to obtain truncated distributions.

In total, the parameters of interest we want to infer given the measured jet scores and di-jet masses consist of: $K = 3$ class fractions $\pi_k$ with the convention that $\pi_1$, $\pi_2$ and $\pi_s$ correspond to $\mathfrak{b}_1$, $\mathfrak{b}_2$ and $\mathfrak{s}$, respectively; two sets of $\{\alpha_j, \beta_j\}$ parameters for each jet type; the exponential rate for the non-resonant di-jet mass $\lambda$ when the event belongs to either $\mathfrak{b}_1$ or $\mathfrak{b}_2$; and the mass mean and standard deviation $\{\mu, \sigma\}$ for the Normal distribution each di-jet mass follows when the event corresponds to $\mathfrak{s}$. These parameters and the probabilistic model they define are depicted as a Graphical Model in Fig. 3.

To quantify and compare the performances of the ABCD method and the Bayesian techniques, we generate different datasets where the unknown we are interested in estimating is

the quantity of signal. To generate pseudo-data, we consider true values:

$$\frac{\pi_1}{\pi_2} = 0.5 \,,$$
$$(\mu, \sigma) = (125, 7) \,,$$
$$\lambda = 0.004 \,,$$
$$(\alpha_c, \beta_c) = (4.8, 7.4) \,,$$
$$(\alpha_b, \beta_b) = (2.9, 1.2) \,,$$

and varying values of $\pi_s$ which along with the $\frac{\pi_1}{\pi_2}$ value and the constraint $\sum_k \pi_k = 1$ fully determine all class fractions. These values are chosen to mimic the qualitative behavior of some of the probability distributions involved in di-Higgs searches. The $\mathcal{S}$ pdfs are chosen by fitting Beta distributions to the reported $\mathcal{S}$ pdfs in Ref. [20] while the mass parameters are similar to what one would expect for a resonant and a non-resonant distribution on the [75, 175] GeV mass range, as detailed e.g. in Ref. [21]. The choice of parameters also achieves a reasonable overlap between signal and backgrounds such that the problem is meaningful.

As discussed in previous paragraphs, each set of parameters has an associated prior. We have chosen parametric forms for the priors, with associated hyperparameters such that a priori the parameters are distributed as

$$\pi_{1,2,s} \sim \text{Dirichlet}(1, 1, 1) \,,$$
$$\mu \sim \mathcal{N}(131.25, 12.5) \,,$$
$$\sigma \sim \mathcal{N}(6.3, 0.7) \,,$$
$$\lambda \sim \text{Uniform}(0.00004, 0.02) \,,$$
$$\alpha_c \sim \mathcal{N}(5.2, 0.52) \,,$$
$$\beta_c \sim \mathcal{N}(7.0, 0.7) \,,$$
$$\alpha_b \sim \mathcal{N}(2.7, 0.27) \,,$$
$$\beta_b \sim \mathcal{N}(1.3, 0.13) \,.$$

The priors and their hyperparameters are meant to reflect a slight mismodelling of the parameters with reasonable associated uncertainty. This uncertainty allows us to assess if the algorithm improves our knowledge about the probability of the true values, which is one of our main interests. We have verified that as long as the prior does not forbid the true values, the specific prior hyperparameters are not very impactful because the number of events is sufficiently large. Observe that using priors whose means are shifted from the true values seeks to mimic the expected real case that the Monte Carlo generator is not perfectly tuned to the data,

We generate different datasets with $N = 20k$ total events where $\pi_s = 0\%$, $0.5\%$, and $1\%$, three reasonable benchmarks where the number of events is high enough for VI to be useful but low enough for finite statistics to provide a relevant limitation. Once inference has been performed, we have access to the MAP for each one of the inferred parameters and therefore to the curves and/or values that they represent in the model. The score and mass generated distributions for a particular dataset can be seen in Figs. 4 and 5, where they are compared to the learned distributions when performing VI. We observe how the MAP distribution is an accurate approximation of the true distribution.

Each event has six numbers, corresponding to the four jet scores and the two masses. That is, the data is six-dimensional, and impossible to fully visualize. However, the data as processed for the ABCD method is two-dimensional and it is instructive to plot it in its own ABCD framework. The observables in the ABCD method consist of the number of $b$-tagged jets and $\chi = \sqrt{(m_1 - m_{h_1})^2 + (m_2 - m_{h_2})^2}$; where in this work for simplicity $m_{h_{1,2}} = m_h = 125$ GeV.

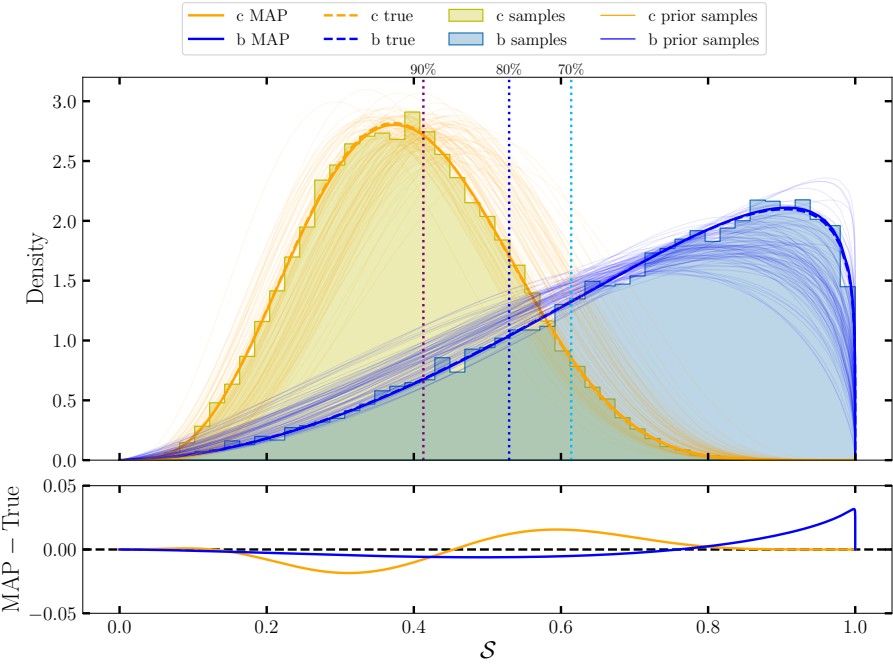

Figure 4: Top: Data distribution of $b$-tagging score values $\mathcal{S}$ for each of the jet types. True (MAP) distributions are shown in dashed (solid) lines for each jet type, while several distributions sampled from the prior for each individual type are shown in thin solid lines. (Dashed and solid lines have large overlapping.) The MAP distributions are inferred from a dataset with $\pi_s = 1\%$. The dotted vertical lines correspond to the WP thresholds we use in this work. Notice that data is four-dimensional in the $b$-tagging scores, but here we project it to one-dimension for the sake of showcasing the inference on the individual jet types. Bottom: Difference between MAP and true distributions for each of the jet types.

The number of $b$-tagged jets is obtained by selecting a working point (WP) $\mathcal{S}_{\mathrm{WP}}$ such that the jet is $b$-tagged if $\mathcal{S}_i \geq \mathcal{S}_{\mathrm{WP}}$. In Fig. 6 we show an example of a sampled dataset classified into $3b$ and $4b$ (where in this context $b$ refers to $b$-tag and not jet type) and $\chi$ divided in signal region SR ($\chi < 25$ GeV) and control region CR, $25$ GeV $\leq \chi < 50$ GeV. The WP is selected such that the True Positive Rate or fraction of accepted true $b$-jets is a fixed number. We plot a few WPs in Fig. 6 to transmit how the different scenarios alter the ABCD hard cut regions.

We emphasize here that this projection and selection of SR and CR is only necessary to obtain a signal estimation with ABCD. When computing the MAP values with VI, we only restrict ourselves to the aforementioned mass window $m_{1,2} \in [75, 175]$ GeV and make no constraints on the number of $b$-tagged jets and the value of $\chi$. This is precisely one of the main advantages of Bayesian Inference as detailed in Section 2. In Fig. 6 we also show the expected dataset distribution, which is obtained by sampling a very large dataset of 10M events and scaling back the event counts to 20k. We observe how fluctuations in the CR will result in noisy estimates of the backgrounds that result in a larger statistical uncertainty on the predicted signal. The Bayesian modelling restricts the possible shape of the distributions and thus reduces the statistical error, at the expense of a possible increase of the modelling error. In this toy problem, because the model is perfectly specified, modelling error is not an issue.

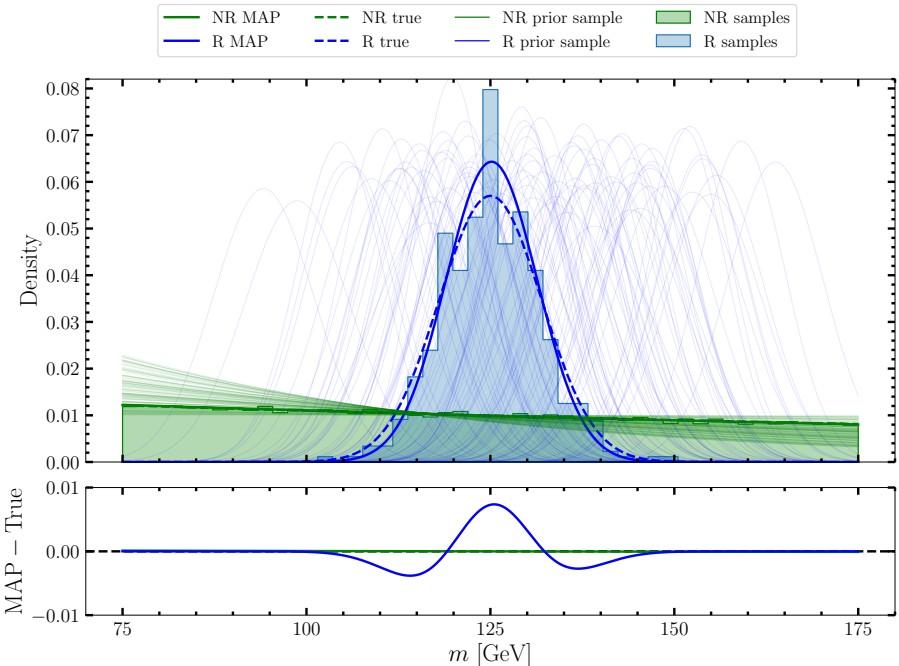

Figure 5: Top: Data distribution of mass values *m* for each of the individual mass types. True (MAP) distributions are shown in dashed (solid) lines for each mass distribution types, while several distributions sampled from the prior for each type are shown in thin solid lines. The MAP distributions are inferred from a dataset with $\pi_s = 1\%$. R and NR stand for resonant and non-resonant, respectively. Bottom: Difference between MAP and true distributions for each of the mass types.

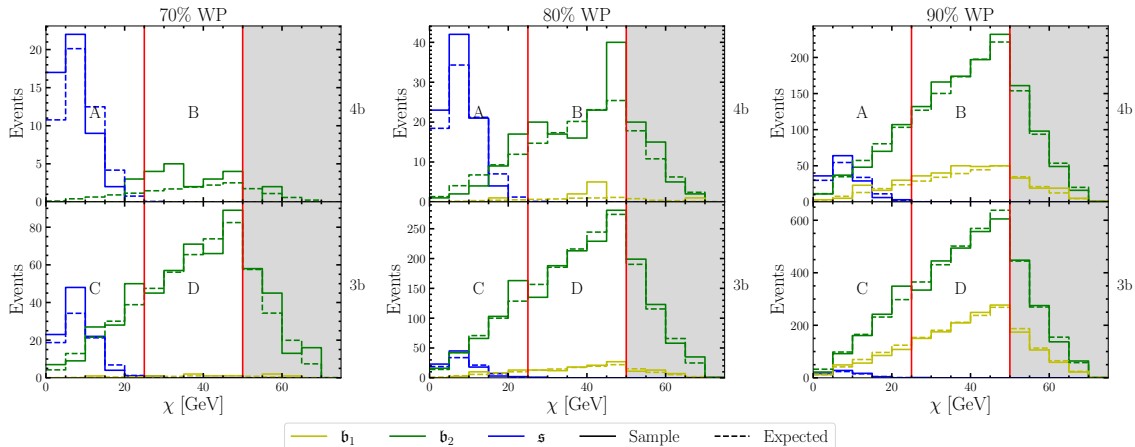

Figure 6: For each one of the used WP (in each column), we plot for a sampled dataset (solid) the $\chi$ distribution in each one of the ABCD regions for each one of the classes (signal and both backgrounds). The corresponding expected distributions, with no fluctuations, are depicted through dashed lines. Notice the different vertical scales, and observe the data migration between the regions as the WP is varied.

## 3.2 Results

Here we detail the results with a small number of signal events and with background only. The results with signal, which consists of several datasets with varying amounts of signal, aim to showcase the power of the model while the results without signal showcase its robustness against false positives.

To evaluate the performance of the model, we first perform a visual test as to whether the learned score and mass distributions are within statistical uncertainties of the underlying distributions. This ensures that the model is actually identifying the proper classes.[5] This is further reinforced by our quantitative metric of choice: the predicted signal events by the model, both in the whole dataset and in the signal region. While for ABCD these are one and the same due to its assumption of signal localization, our Bayesian model does not assume such a strong localization. As a matter of fact, the Bayesian framework provides the tools to solve the problem of having non-negligible signal in the control region in comparison to signal region, a problem also addressed with different tools in Ref. [14]. Another strength of the strategy is that to obtain the predicted number of signal events we can use a soft-assignment strategy where we compute for each event the probability

$$p(\mathbf{z}_n = \mathfrak{s}|\mathbf{x}_n, \theta^{\text{MAP}}) = \frac{p(\mathbf{x}_n, \mathbf{z}_n = \mathfrak{s}|\theta^{\text{MAP}})}{p(\mathbf{x}_n|\theta^{\text{MAP}})}, \tag{19}$$

and obtain the estimated number of signal events $S_{\text{pred}}$

$$S_{\text{pred}} = \sum_{n=1}^{N} p(\mathbf{z}_n = \mathfrak{s}|\mathbf{x}_n, \theta^{\text{MAP}}). \tag{20}$$

This is different from a hard-assignment strategy, where events are assigned the class label for which Eq. 19 is maximized. A soft-assignment strategy is preferred to be consistent with the probabilistic nature of the Bayesian methodology, and ensures that we preserve the contributions of the full dataset to $S_{\text{pred}}$.

### 3.2.1 Small signal

We consider two different (small) signal fractions, $\pi_s = 1\%$ and $0.5\%$. Since we are using a point-estimate VI algorithm, to study the robustness of the method and assess the uncertainty of the signal predictions we sample 25 different datasets for each $\pi_s$. These runs can be used to assess the uncertainty on a given quantity (in our the case the ratio between predicted and true signal events) by computing its mean and the standard deviation of said mean. Each dataset has a different amount of signal events which yields a noisy estimate of $\pi_s$ with its corresponding statistical information. We show in Figs. 4 and 5 the resulting distributions from a particular example with $\pi_s = 1\%$, where it is seen how the true distributions are well captured by the inference.

To compare both methods, ABCD and Bayesian inference on a mixture model, we compare predicted and true signal events in each method. We do this in two different sets, one restricted to the signal region A, and the other in the full dataset. The first approach pursues to compare both methods within the rules of the ABCD method, in which the hypothesis is that the signal is localized in A. Since this assumption is usually not fully accomplished, we also study the second approach which is more relevant for observing signal events in general and can highlight the known ABCD pitfalls when small signal fluctuations bias the background estimation. We show

---

[5]In a real case scenario, without access to the true values, one tests the modelling by sampling replicates of the data using the model with the inferred parameters, and computing the compatibility of the ensemble of replicates with the real data [19, 21].

both approaches for the three WP=70%, 80% and 90% in each plot in Fig. 7, where we show the predicted signal events and the difference between predicted and true signal events as a function the true number of signal events in the corresponding region.

To interpret Fig. 7 with respect to the ABCD method, one should take into account how the signal and background events are redistributed among the regions in Fig. 6 as the WP is increased. For a tight[6] WP (70%) we have that region A consists of almost all signal events with very few background events, however, at the price of leaving too many signal events in region C. Since the background in A is very small for this tight selection, any relative errors in background estimation due to contaminating signal events in C does not greatly affect the signal estimation in A. This is because the signal estimation is based on the total number of events in A minus background estimation in A and the former is much larger than the latter. Henceforth, the signal estimation for A with a tight WP is very good in the left column in Fig. 7, but at the price of a very bad estimation of the total number of signal events in the sample, as depicted in the right column in the same figure. Observe that one can easily understand the observed lower bias in the signal estimation in A because of the signal contamination in C.

As one increases the WP, there is a migration of signal events from C to A which increases the localization of the signal, but also many more background events enter into all the regions since there are more $b$-tagged jets. The rate of population growth is different for background than for signal because the increase of false-positive $b$-tags from the non-b jets is larger than the increase in true-positive from the true b-jets. In particular, the background in A increases noticeably more than the signal in A. Therefore, although there is a smaller relative bias in the background estimation in A –because of less contaminating signal events in C–, the increase of background relative to signal in A yields a more biased signal estimation in A. This can be seen in both columns in Fig. 7, and we have verified that the signal contamination in C is generating this bias. In any case, a looser WP (90%) yields a better estimation of the total number of signal events (right column in figure) due to better localization.

The notorious larger spread in the signal estimation as the WP increases is due to the same difference in growth between background and signal that biases the signal estimation. For tight WPs, the signal estimation is mostly the population in A corrected for a small background. Thus, the error in the signal estimation is mostly the usual Poisson error. The error in the ratio is thus approximately obtained by propagating the ratio between two Poisson variables, the signal estimation and the true signal. For larger WPs, because the signal-to-background fraction is smaller the signal estimation is driven by the background estimation and its subtraction from $N_A$. Thus, the error in the signal determination is more dominated by the errors in the background estimation. Because the errors in the background estimation are larger than the Poisson errors of each individual measurement, the uncertainty on the signal estimation increases noticeably more than the uncertainty on $N_A$ and on the true signal which also increase due to the increased WP. The uncertainty on the ratio between signal estimation and true signal increases accordingly.

As it can be seen from the discussed figure, in all scenarios there is a better estimation from Bayesian inference than using the ABCD procedure. Because Bayesian inference does not rely on a selected WP it is therefore not affected by any problems associated with it. Of course, in the present work this is achievable thanks to the simplification in the parameterization of the $b$-tagging curve score and a more realistic case is discussed in Section 4. Also observe that the left plot in Fig. 7 has a dependence on the WP for Bayes solely because the signal region A has a dependence on it.

We summarize the above results in Fig. 8, where we evaluate the differences between the methods for different WPs and signal fractions by comparing the ratio of predicted-to-true signal events in the signal region A and in the full dataset. Since the ABCD signal prediction is

---

[6]It is referred as *tight* because it practically does not tag non-b jets as b.

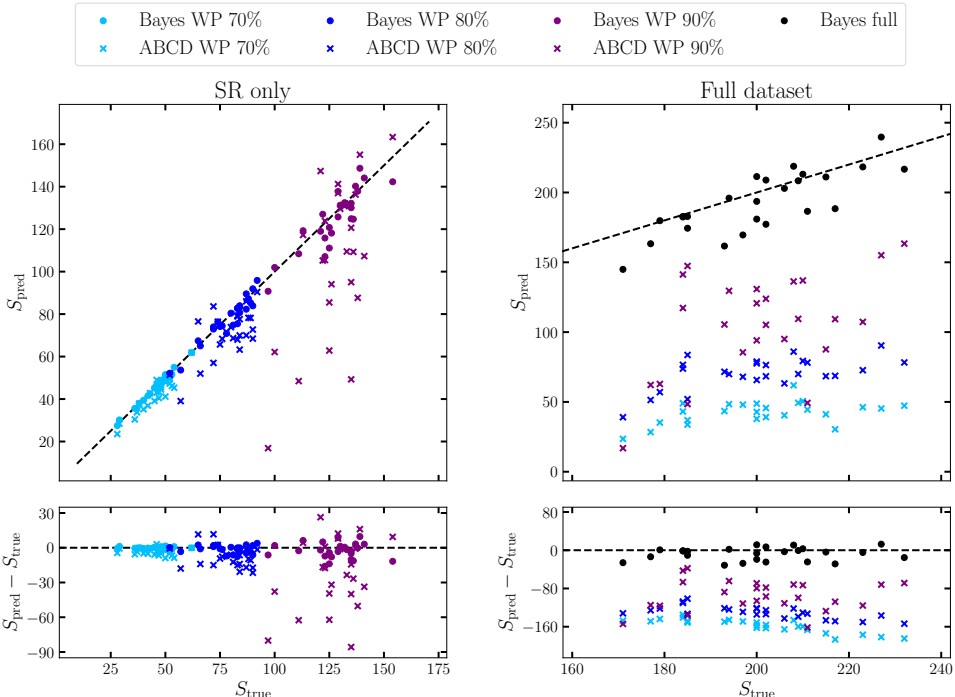

Figure 7: Top (bottom): Predicted signal count (difference between predicted and true signal counts) as a function of true signal count in the ABCD signal region (left) and in the whole dataset (right). The observed biases and spreads in the ABCD predictions are discussed in the text. Dashed lines represent perfect matching, and we see that in all cases the Bayes framework compares favourably. See discussion in text concerning the ABCD performance when considering both left and right results. (Although Bayes does not depend on the WP, the ABCD signal region on which we are counting the events does depend on the WP. This is why there is a WP label assigned to the Bayesian method as well.)

the same for both the SR and the full dataset, it should yield incorrect ratios for both datasets if the signal localization assumption is not exact. In that case, for the ABCD method the performance degrades for larger signal fractions due to a more evident lack of signal localization. Conversely, the Bayesian strategy is not degraded by the SR region definition nor by changes in the signal fraction. This is evidenced by the fact that the predictions in all six scenarios (three WPs with two signal fraction each) are consistent with the true value within statistical fluctuations.

### 3.2.2  No signal

We now consider the case where no signal is present to assess the robustness of the Bayesian method against producing spurious signals estimations. For the case where no signal is present but we still consider it as part of the probabilistic model, we find that the model is robust in the sense that it will suppress the signal class. This is seen in Fig. 9, where we compare the predicted signal events for the Bayesian method and for the ABCD method in the signal regions defined by three possible WPs and in the full dataset. To obtain the statistical uncertainty of the estimation, we average over 25 runs where each dataset has no real signal events.

Both for ABCD and the Bayes method, we obtain a usually non-integer signal prediction $S_{\text{pred}}$. A more thorough analysis should assess the compatibility of this prediction with the actual presence of signal events, which can only take discrete values. This could be done by

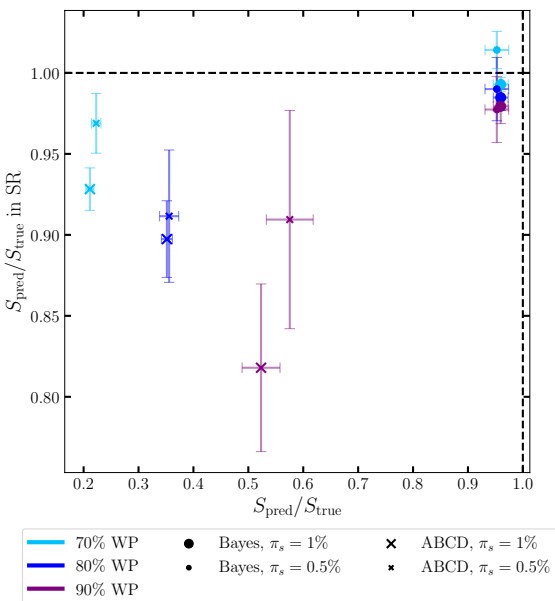

Figure 8: Comparison between signal estimation values for ABCD and Bayes methods, given different $\pi_S$ and WP values. Each point is calculated as the mean of the 25 runs in Fig. 7, and error bars indicate the corresponding standard deviation. See discussion in text about these results.

using a Poisson distribution with rate given by the estimated $S_{\text{pred}}$ and evaluating the probability of zero events. Another possibility is to study the compatibility of the data with the $\pi_s = 0$ hypothesis. This could be done in a fully Bayesian manner by computing the evidence ratio (see Section 3.4 in [15]) between the two probabilistic models (with and without signal), performing a posterior predictive check [22] for each model; or approximately by re-doing the MAP estimation with two classes and computing a model comparison metric such as the Bayesian Information Criterion difference between models (see Section 3.5 in [15]). However, the presented analysis already provides a satisfactory proof of robustness and we leave these more involved model comparison techniques for future work.

For small signals and due to statistical fluctuations in the CR, the ABCD method may predict negative signal events. This could be avoided by setting to zero any negative predictions. However, this has the undesirable effect that the underlying distribution is no longer Gaussian and thus the errors cannot be computed simply by averaging over different runs. The Bayesian method will by definition predict non-negative $S_{\text{pred}}$ through the non-negative fraction $\pi_s^{\text{MAP}}$. However, because it will be very close to zero, a simple estimation of the $S_{\text{pred}}$ uncertainty by computing the standard deviation of a sample of estimators obtained from different datasets may fail to account for the skewed nature of the probability distribution and yield incorrect error bars. In practice, we observe that the variation on the estimate is very small and does not appear to be that sensitive to the hard $\pi_s = 0$ boundary. We consider the symmetric error bars appropriate and representative of the uncertainty on the estimator. We note that this problem would not arise in a fully Bayesian treatment where we compute the full posterior distribution $p(\pi|\mathbf{X})$ and thus can obtain the relevant confidence intervals on $S_{\text{pred}}$ taking full account of all boundary conditions.

From Fig. 9 we observe how the Bayesian method not only is robust against no signal, suppressing the signal class fraction so that it effectively predicts no signal but also avoids predicting negative amounts of signal by definition.

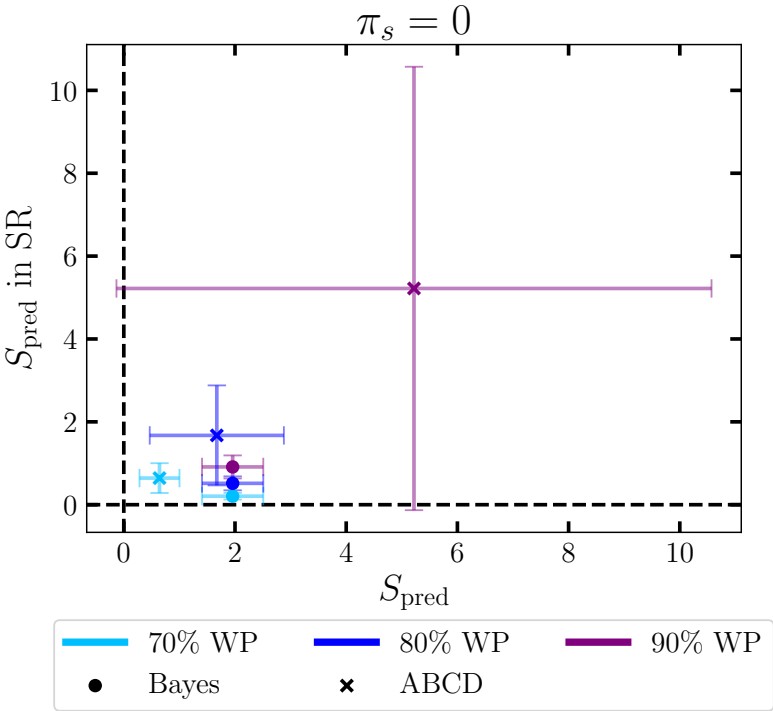

Figure 9: Comparison between signal estimation values for ABCD and Bayes methods, given WP values, for the case of signal absence in the data ($\pi_s = 0$). Each point is calculated as the mean of 25 runs, and error bars correspond to the standard deviation of the mean.

# 4 Outlook

The presented results explore a proof-of-concept for a method that aims to study improvements in the sensitivity of some HEP observables. As such, there are many points which deserve further discussions.

A first point to notice is that, although the above results seem very convincing within the presented framework, there is still a way to go before they can be well established in real scenarios. In the controlled layout where the toy model is discussed, we identify at least two main reasons that lead to the improvement in going from ABCD to a Bayesian framework: *i)* in the proposed example the Bayesian framework has six dimensions (the four jet scores and the two invariant masses), in comparison to two dimensions ($N_b$ and $\chi^2$) used by the ABCD method; and *ii)* The Bayesian framework uses soft-assignments with no hard cuts, which is more powerful than the hard cuts needed for the ABCD method. The first point has to do with exploiting the mutual information at the event-by-event level, which is an aspect sometimes foreseen in experimental analyses and could be indicating room for sensitivity improvement in observables. See Refs. [19, 21, 23] for some discussions, proposals and results in this direction.

A second point to observe is that in the described toy model we perform simulations using the same probability density functions that we use to extract the parameters. This choice ensures that we do not have to correct for any bias in our model and enjoy its benefits regarding reduced statistical uncertainties on the signal fraction. In a more realistic scenario we should use pseudo-data generated from physical Monte Carlo (such as for instance `MadGraph` [24] → `Pythia` [25, 26] → `Delphes` [27], or similar), and then elaborate a model that can still capture the inner structure of the data. In this direction, in Ref. [28] it is studied how to replace a simplistic Beta for the *b*-tagging score distributions with arbitrary continuous functions, as in

real scenarios [20, 29, 30]. A next stop in the way for a more realistic scenario is to merge both studies and analyze its scope. This study shall be performed using the full Bayesian posterior, instead of point estimate, which should improve its performance.

One of the objectives of this work, in addition to its standing-alone results, is to provide a building-block for a more ambitious enterprise that is to propose a Bayesian-based analysis for the process $pp \rightarrow hh \rightarrow b\bar{b}b\bar{b}$ at the LHC. The results in this work are along this direction. Once within this enterprise, one can envisage many other building-blocks that should be achieved before the question of whether a Bayesian framework can improve the $pp \rightarrow hh \rightarrow b\bar{b}b\bar{b}$ sensitivity could be correctly formulated. Among these, one should include the treatment of experimental systematic uncertainties in the inference, with a special focus on those that break our modelling assumptions. In particular, conditional independence can be broken by systematic-induced dependence between the different jets. These relations, which could be type- and class-dependent, can relate scores and masses in unforeseen ways. These so-called nuisance parameters capturing relevant physics thus have to be addressed by modifying the probabilistic model accordingly.

One should also perform a study adding the real backgrounds, which include single Higgs production ($pp \rightarrow hjj, hbb$), fakes from conversions, $t\bar{t}$, non-resonant $4b$, etc., being the latter one of the most challenging because of its abundance and irreducibility in what has to do with the particles identification. Including many backgrounds increases the number of parameters of the model, and it should be done exploiting prior knowledge from these processes to avoid a drawback in the sensitivity performance. Another feature to add when including many backgrounds is to simultaneously compute the appropriate jet combination to determine the two Higgs candidates with all other parameters in the model. In general, all these objectives constitute a very difficult and costly undertaking but it is worth it for the objective pursued. We expect to produce more contributions along these lines in the near future.

In a still more ambitious plan (chimera?), one could consider merging different ingredients of the previous program together into a unique Bayesian analysis, and hence improving the overall performance. For instance, the multivariate analysis used to assign $b$-tagging scores to the different jets is performed separately to the whole $pp \rightarrow hh \rightarrow b\bar{b}b\bar{b}$ analysis. However, since we have the prior knowledge of the backgrounds involved in the analysis, we can frame the learning of the proper tagger as a Bayesian neural network to be combined with the mixture model.

## 5 Conclusions

Motivated by a potential room for improvement in the ABCD method of background estimation we have proposed to perform Bayesian inference on a *mixture model* to analyse a set of data containing a signal and several backgrounds. The aim of the proposal is to leverage the mutual information of the observables at the event-by-event level. The method presented here implements standard Bayesian inference techniques to analyse data arising from a signal and several different backgrounds, with an arbitrary number of independent observables measured in each event, with no need of control regions. It naturally allows to implement *soft-assignments* to the events, via their probability to belong to the different processes. By comparison, the standard ABCD method is restricted to two independent observables with two rigid selections for each, as well as the need of a control region, among other limitations.

As an example, we have considered a toy problem inspired by the di-Higgs production process: $pp \rightarrow hh \rightarrow b\bar{b}b\bar{b}$. We studied a set of simplified pseudo-data mimicking events from $hh \rightarrow b\bar{b}b\bar{b}$, as well as the QCD backgrounds $b\bar{b}c\bar{c}$ and $c\bar{c}c\bar{c}$. As observables we have

considered the *b*-tag score of each jet and two invariant masses arising from jet-pairing. For simplicity we have modeled the score probability distribution of the jets with Beta functions and the invariant mass of di-jets with truncated normal distributions and a decaying truncated exponential for signal and backgrounds, respectively. We have considered different fractions of signal of percent order, as well as the case of background only. For this toy example we have compared the performance of the ABCD method and Bayesian inference on the *mixture model*, finding that the latter gives better estimations of the number of signal events and smaller errors than the former one, both in the signal region in ABCD and in the whole dataset. A particular advantage for the Bayesian framework, is that it is not affected if signal events lie outside the ABCD signal region. A detailed discussion about this comparison can be found in Section 3.2.

Our implementation is far from a realistic analysis for several reasons: we considered only two backgrounds, excluding for example the irreducible $4b$ QCD production; we generated the data and inferred the parameters of the distributions with the same density distribution functions; we used very simple functions that are not realistic; we did not include full experimental systematic uncertainties among other issues. Our simplistic analysis of the toy model must be read as a first step towards a realistic analysis of di-Higgs production decaying to $4b$, and we conceive it as a start of a more ambitious program that aims to address the mentioned issues, reaching a realistic description of different physical processes of interest in colliders.

Finally, one should observe that from the results and description in the present article, the Bayesian inference techniques represent a different paradigm than the ABCD method for background estimation. In this sense, it would be very interesting to tackle other processes of interest with the proposed method, such as for instance $ee \rightarrow Zhh$, $pp \rightarrow X \rightarrow VV \rightarrow 4\ell, 2j2\ell, 4j$, which should benefit because of the many independent and approximately independent observables in the final state.

# Acknowledgments

EA and MS are grateful to participants of the Workshop Voyages Beyond the SM V for fruitful discussions. EA thanks R. Piegaia for useful discussions.

**Funding information** EA, LD, AS and ST acknowledge support from CONICET PIP-11220200101426 and FONCyT PICT-2018-03682, MS acknowledges support in part by the DOE grant de-sc0011784 and NSF OAC-2103889.

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
