# Peer review of "Improvement and generalization of ABCD method with Bayesian inference"

_SciPost Physics Core, doi:SciPost Phys. Core 7, 043 (2024)_

## Round 1 · Referee Report · Anonymous (Referee 1) · 2024-4-5

Strengths

A novel method for estimating background in particle physics collision analysis.
A promising method potentially improving existing ones.
It proven to work in more general cases it could improve the reach of many analyses in particle physics

Weaknesses

The method is not proven to work in a general case.
The proof is based on a oversimplistic example
The clarity of the article is improvable.

Report

The article presents an interesting and novel method that could be useful for the background estimation in a large variety of analyses at LHC, hence extending its physics reach. However, its applicability to a realistic case is not proved, only presenting its operation in a very simple example where other methods (simpler and probably more effective) will also work. The assumption of totally independent variables with known pdfs is very strong and under that hypothesis several alternatives are possible. In fact, a direct ML fit could be applied under those assumptions and is known to be optimal from the point of view of statistics.
On the contrary, this method is likely to suffer less from variable dependence and hence it is promising, this point must be proved. In general, if possible, or at least with a more realistic case incorporating correlations.
It is also not clearly stressed how to define priors in a general case and how the results depend on priors on the described the example.
Finally, it would be interesting to have a clearer and self-consistent description of the method, without having to go to external references.
Regards

Requested changes

There are three major things that should be envisaged.
1-Extend the example to evaluate the performance in the presence of correlations between variables.
2-Explain how priors can be defined in general and how they affect the result in the shown example
3-Improve the description of the method
Some additional changes would improve significantly the quality of the article:
- “tune down” some statements in the introduction. For example, you seem to say that ABCD is critical for H to 4b, while it is just one of the challenges (and there are alternatives)
- After eq (1) “observables have background distributions” is not correct
- You have “Introduction” in uppercase in the middle of a sentence
- Don’t understand the sentence “As it can be appraised, the ABCD method is very clever and simple, and it has worked with excellent results and achievements in HEP, as discussed in the Introduction. However, if the hypotheses are not exactly satisfied, the predictions would deviate from their corresponding true values, as expected. Not only that, but also if the total number of events in B, C or D is small, then its Poisson fluctuation would propagate to the signal and background events predicted in A.”, that is basically also true also for the proposed method, isn’t it?
- Description of eq 2 and the notation is confusing. For example theta is both used for all parameters or excluding pi; it is not not terribly clear which are vectors and which scalars
- Eq (4) is confusing, what means the sum over delta? Aren’t you assuming p(z_n)=1? Not clear how you get from (4) to (5), that seems the right one.
- Explain the basics rather than ask the reader to Ref[15] ( a full book).
- Fig 7 and others, A figure of the difference between true and measured will permit to better judge the spread and bias, that 2D.
- Fig 8, not sure if it tells much, justify. In particular, why confront the methods for the whole region when ABCD method is not designed for that? If you want to estimate the signal in the whole region, you’ll do a global fit, don’t you?

  • validity: good
  • significance: good
  • originality: high
  • clarity: good
  • formatting: good
  • grammar: good

Author:  Santiago Tanco  on 2024-05-31  [id 4532]

(in reply to Report 1 on 2024-04-05)

Dear Reviewer,

Please find our response to your report in the attached file, along with the manuscript where all changes are highlighted in blue.

Best regards,
Santiago Tanco on behalf of all the authors

Attachment:

ABCD_Bayes.pdf

Author:  Santiago Tanco  on 2024-05-27  [id 4520]

(in reply to Report 1 on 2024-04-05)

We thank the reviewer for the very thorough report. We have prepared a response addressing all of her/his comments, along with a new version of the manuscript that includes the requested changes. We asked the editor a few days ago how to send the new version through the channels established by the journal. In any case, and in order to not lose the conversation's momentum, if we don't receive these instructions we will reply to the reviewer with our response and the new version of the draft through this conversation.

Best regards

---

## Round 2 · Referee Report · Anonymous (Referee 1) · 2024-6-12

Report

I believe the responses properly address all my questions and comments and the draft has interesting results. I hence recommend its publication.

Regards

Recommendation

Publish (meets expectations and criteria for this Journal)

---

## Round 2 · List of Changes

Following the report of the reviewer, we have made changes to the manuscript that are explicitly shown in blue.

---

## Editorial Decision

published